# Efficacy and safety of anti-viral therapy for Hepatitis B virus-associated glomerulonephritis: A meta-analysis

Baohui Fu[1☯], Yue Ji[1☯], Shouci Hu[2], Tong Ren[1], Maheshkumar Satishkumar Bhuva[3], Ge Li[4]*, Hongtao Yang[1]*

**1** Department of Nephrology, First Teaching Hospital of Tianjin University of Traditional Chinese Medicine, Tianjin, China, **2** The First Affiliated Hospital of Zhejiang Chinese Medical University, Zhejiang Provincial Hospital of Traditional Chinese Medicine, Zhejiang, China, **3** International Department, Tongji University School of Medicine Affiliated Shanghai Pulmonary Hospital, Shanghai, China, **4** Public Health Science and Engineering College, Tianjin University of Traditional Chinese Medicine, Tianjin, China

☯ These authors contributed equally to this work.
* tjpdyht@163.com (HY); ligeself@163.com (GL)

**Data Availability Statement:** All relevant data are within the manuscript and its Supporting Information files.

**Funding:** This work was funded by the National Natural Science Foundation of China (Research on

# Abstract

## Objectives

To assess the potency of anti-viral treatment for hepatitis B virus-associated glomerulonephritis (HBV-GN). Method: We searched for controlled clinical trials on anti-viral therapy for HBV-GN in MEDLINE, Embase, the Cochrane Library, and PubMed from inception to March 11th 2019. Seven trials, including 182 patients met the criteria for evaluating. The primary outcome measures were proteinuria and changes in the estimated glomerular filtration rate, and the secondary outcome measure was hepatitis B e-antigen clearance. A fixed or random effect model was established to analyze the data. Subgroup analyses were performed to explore the effects of clinical trial type, anti-viral drug type, age, and follow-up duration.

## Results

The total remission rate of proteinuria (OR = 10.48, 95% CI: 4.60−23.89, $I^2$ = 0%), complete remission rate of proteinuria (OR = 11.64, 95% CI: 5.17−26.21, $I^2$ = 23%) and clearance rate of Hepatitis Be Antigen (HBeAg) were significantly higher in the anti-viral treatment group than in the control group (OR = 27.08, 95% CI: 3.71−197.88, $I^2$ = 63%). However, antiviral therapy was not as effective regarding the eGFR (MD = 5.74, 95% CI: -4.24−15.73). In the subgroup analysis, age and drug type had significant impacts on proteinuria remission, and study type and follow-up duration only slightly affected the heterogeneity.

## Conclusion

Antiviral therapy induced remission of proteinuria and increased HBeAg clearance but failed to improve the eGFR. Pediatric patients were more sensitive to antiviral therapy than adults. IFNs seem more effective but are accompanied by more adverse reactions than NAs.

modernization of traditional Chinese Medicine: Major Research Program), Grant/Award Number: 2019YFC1709400. The funder had no role in study design, data collection and analysis, decision to publish, or preparation of the manuscript.

**Competing interests:** The authors have declared that no competing interests exist.

# Introduction

Despite the availability of potent antivirals and vaccination against the hepatitis B virus (HBV), chronic HBV infection remains a significant socioeconomic burden. The global prevalence of chronic HBV in 2016 was close to 3.9%, which equates to approximately 292 million HBsAg-positive individuals, making it one of the most common human pathogens, and there is large regional variation[1]. In addition to its effects on the liver, patients with HBV have extrahepatic manifestations of renal disease. In a single-center retrospective trial, Zhang et al[2] reported that 3% (n = 352) of a total of 11,618 kidney biopsies performed from 1987 to 2012 in Beijing, China, were HBV-GN positive.

Treatment of HBV-GN remains a challenge for clinicians, the use of steroids or immunity inhibitors is still controversial because of the risk of activating viral infections, so the mainstay of treatment continues to be antiviral therapy. Several drugs are now available for the treatment of chronic HBV infection and are mainly one of two types: nucleos (t) ide analogs (NAs), such as lamivudine, adefovir, entecavir, telbivudine, and tenofovir, and interferons (IFNs)[3]. However, their efficacy and safety remain poorly established and uncertain.

Several meta-analyses[4–10] have been published regarding the effects of different therapies in HBV-GN patients; three studies[4,6,7] focused on the effects of antiviral treatment on HBV-GN but were completed before 2011. However, previous investigators did not provide details regarding to therapies, such as treatment for patients of different ages, antiviral drug selection or treatment duration. Standardized treatment guidelines that are personalized and appropriate for the current therapeutic regimen are lacking. Therefore, we performed an updated systematic review of clinical trials to evaluate the efficacy of antiviral therapy for HBV-GN.

# Methods

We performed our meta-analysis as suggested by the Cochrane guidelines[11]. The PRISMA (Preferred Reporting Items of Systematic reviews and Meta-Analyses) checklist was used to standardize the details[12], and the PRISMA checklist is shown in S1 Table. The protocol and registration information are available at http://www.crd.york.ac.uk/PROSPERO/ (registration number: CRD42019125268).

## 1. Search strategy

We searched PUBMED, MEDLINE (Ovid Online), Embase and the Cochrane Library from inception to March 11th 2019. We also searched for related terms using the medical subject headings (MeSH) database. The key words were "interferon", "nucleoside analogs", "anti-viral", "lamivudine", "adefovir dipivoxil", "entecavir", "telbivudine", "hepatitis B virus", "glomerulonephritis", and their synonyms and related terms. We also manually checked the reference lists of nephrology textbooks, review articles and reports from academic congresses.

## 2. Eligibility criteria

Trials were selected based on the following inclusion criteria:

1. Randomized controlled trials (RCTs) and cohort clinical trials (CCTs) in which the intervention was antiviral therapy (IFN or NAs) were included. All studies had a follow-up period of at least 6 months and had been published in a journal.

2. All included patients were diagnosed histologically based on the existence of HBeAg, HBcAg, or HBV-DNA in renal biopsy specimens, accompanied with HBV surface antigen

(HBsAg)-positive serum and were free from lupus nephritis and other secondary glomerular diseases[13].

Clinical trials meeting the following criteria were excluded:

1. Self-control studies or case reports.

2. Patients who were diagnosed with glomerulonephritis but not secondary nephritis caused by the hepatitis B virus.

## 3. Selection of studies and data extraction

All articles identified in the search were screened independently by two authors (Baohui Fu and Yue Ji) and deemed eligible if they met all inclusion and exclusion criteria. Any discrepancies were resolved by an expert (Shouci Hu). The references from all eligible articles were screened and included using the same process as above.

Data on the study (year, country, authorship, journal, study design and number of participants), demographic characteristics (glomerulonephritis status, age, sex, ethnicity, and pathological type of renal biopsy), intervention details (provider, doses and modalities of treatment, placebo group set up, duration of intervention, follow-up interval and follow-up duration), and outcomes (including laboratory reports: complete remission (CR) and partial remission (PR) of proteinuria; estimated glomerular filtration rate (eGFR); HBeAg clearance; adherence status and adverse drug reactions) were extracted from eligible articles independently by Baohui Fu and Yue Ji.

## 4. Risk of bias assessments

Two independent reviewers (Baohui Fu and Yue Ji) evaluated the risk of bias, when they disagreed, a third author (Shouci Hu) was involved; the three reviewers discussed the article until all reviewers reached a unified conclusion. We evaluated the quality of the RCTs with the Cochrane Collaboration's tool for assessing risk of bias for RCTs[11], and the cohort studies were evaluated with Newcastle−Ottawa scales[14].

## 5. Statistical analysis

Treatment effects were calculated as odds ratios (ORs) with 95% confidence intervals (CIs) to assess dichotomous outcomes. The mean differences (MDs) were used to evaluate the differences of continuous data. The standard deviation (SD) of changes were calculated according to the formula in Cochrane investigator's handbook 5.1.0, the value of correlation coefficient (Corr) was 0.5[11]. The heterogeneity test was performed using the chi-square test ($\chi^2$) and the $I^2$ statistic, where $I^2 > 50\%$ indicated significant heterogeneity[15]. In the presence of statistically significant heterogeneity ($I^2 > 50\%$), the Mantel-Haenszel method in the random-effect model was selected; if $I^2 < 50\%$, the fixed-effect model was used for the meta-analysis[16]. A sensitivity analysis or subgroup analysis was used to discover heterogeneity. Publication bias was assessed by Egger's test and Begg's test. All statistical analyses were performed using RevMan 5.3 (Cochrane Collaboration, Oxford, England). The sensitivity analyses and publication bias assessment were performed using STATA 13.1 (Stata Corp., College Station, Texas, USA). When original data were not mentioned in the article but were found presented in reports and references as functional X-Y type scatter or line plots, Plot Digitizer 2.6.8.0 was used to digitize scanned plots of the functional data. Subgroup analyses were planned to explore potential sources of variability in the observed treatment effects. When possible, subgroup analyses were performed for clinical trial types, antiviral drug types, age and follow-up duration.

## 6. Study outcome measures

The primary outcomes that this systematic review focused on were renal remission rate and estimated glomerular filtration rate (eGFR), and the secondary outcome measure was HBeAg clearance. Complete remission (CR) of proteinuria was defined as urinary protein excretion less 0.3 g in 24 hours. Partial remission (PR) represented a decline in urinary protein excretion by 50% or a decline below the baseline value with an absolute amount greater than 0.3 g in 24 hours. Virologic response was defined as clearance of HBeAg from serum. These definitions are the standard definitions used in the current scientific literature.

# Results

## 1. Summary of included studies

We identified 334 potentially eligible records according to the search strategy. The selection process is illustrated in Fig 1. The titles and abstracts of these records were screened for inclusion, and 66 studies met the inclusion criteria. Based on our exclusion criteria, 56 of these studies were excluded after the full-text screening (Fig 1).

Finally, seven trials, involving 182 participants, were included in our study. Six of the articles were CCTs [17–22], and one was an RCT [23]; they included 44 females and 138 males. In addition, all the HBV-GN patients had kidney biopsies; membranous nephropathy (MN) was found in 134 patients (73.6%), followed by IgA nephropathy in 24 (13.2%), membranoproliferative glomerulonephritis (MPGN) in 12 (6.6%), focal segmental glomerulosclerosis (FSGS) in four, other pathological types, such as mesangial proliferative glomerulonephritis (MSPGN) and minimal change nephropathy (MCGP), in two patients and IgM nephropathy in one patient. Two patients had combined lesions, with one having a combination of MN and MPGN and the other having FSGS and IgAN. The characteristics are described in Table 1.

## 2. Risk of bias assessments

The only RCT[23] in our analysis was low risk in terms of random sequence generation, blinding of participants and personnel, blinding of outcome assessment and incomplete outcome data but had an unclear risk with respect to allocation concealment, selective reporting and other bias (S2 Fig). The biases of the observational cohort studies, evaluated by the Newcastle−Ottawa scale, are shown in Table 2. Three studies had seven to eight stars, which indicates very good quality; four studies had five to six stars and are regarded as being of good quality. After reading the clinical trial designs and protocols of the included literature in detail, no reporting biases were found.

## 3. Effects on proteinuria remission

**3.1 Subgroup analysis of the association between efficacy and study type.** Seven studies, involving 182 patients, evaluated the efficacy of antiviral therapy for HBV-GN. We evaluated the CR and total remission (CR+PR) rates after treatment with a fixed-effect model (Fig 2). There was no clinical and statistical heterogeneity in total remission of all trials ($I^2 = 0\%$, $P = 0.80$), and the test for the overall effect was significant ($P<0.00001$). The total remission rate of proteinuria was obviously higher in the antiviral treatment group than in the control group in all studies (OR = 10.48, 95% CI: 4.60−23.89). The antiviral treatment significantly affected CR in all trials (OR = 11.64, 95% CI: 5.17−26.21). There was no heterogeneity between data sources in the CR of all trials ($I^2 = 23\%$, P = 0.26). The $\chi^2$ test of heterogeneity was not significant in the CR evaluation from the CCTs ($I^2 = 9\%$, $P = 0.36$, OR = 8.47, 95% CI: 3.67

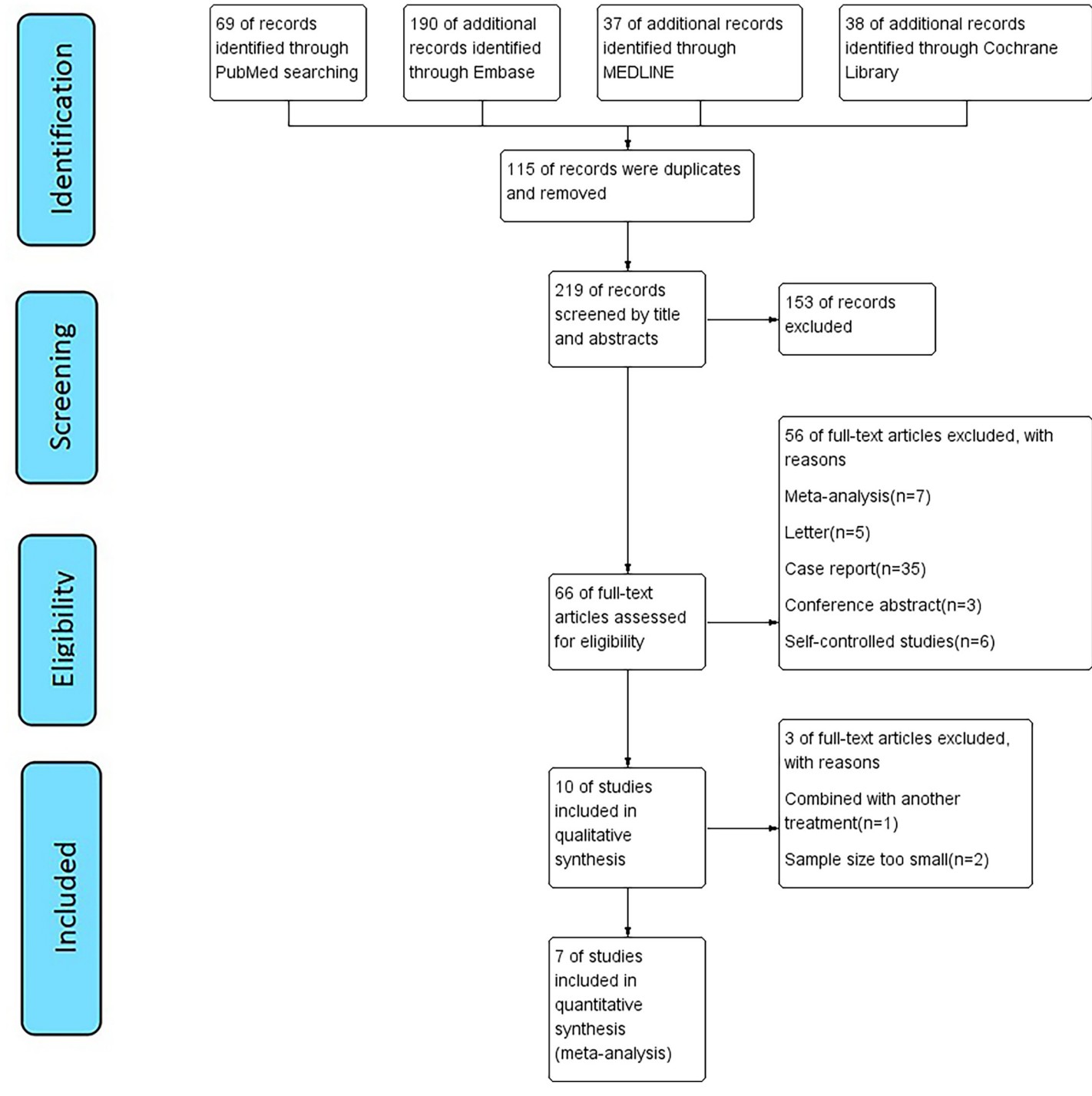

**Fig 1. Flow chart of study selection.**

−20.80). We conducted a sensitivity analysis of all studies (S3 Fig). Our analysis determined that the study by Sun's study[17] and Panomsak's study[20] was the source of heterogeneity.

**3.2 Subgroup analysis of the association between efficacy and antiviral drug type.** To compare the effects of IFNs and NAs, we analyzed two subgroups. NA treatment for HBV-GN was assessed in 4 studies, and IFN treatment was assessed in 3 studies. Panomsak's[20] study

**Table 1. Characteristics of the included trials.**

| Study | Year | Region | Study design | Mean age (y) | Sex | Intervention | Control | Duration | Follow-up | Drop out | Outcomes |
|---|---|---|---|---|---|---|---|---|---|---|---|
| Sun et al[17] | 2017 | Shanghai | CCT | 42.8 ±13.2 | M:28 F:10 | LAM (100 mg/day orally)+ACEI/ARB (n = 20) | ACEI/ARB alone (n = 18) | 12 mo | 12 mo | 0 | ①② |
| Bhimma et al[18] | 2002 | Durban, South Africa | CCT | 8.9 | M:34 F:5 | IFN-α2b (10 million units/m2 3 times/week)(n = 19) | control of edema and hypertension (n = 20) | 16 weeks | 40 weeks | 5 | ①② |
| Tang et al [19] | 2005 | Hong Kong, China | CCT | 45.5 ±19.0 | M:14 F:8 | LAM 100mg/d+ACEI or ARB (n = 10) | ACEI/ARB (n = 12) | NA | 49.2±16.5 mo | 0 | ① |
| Panomsak et al[20] | 2006 | Thailand | CCT | 40.9 | M:10 F:7 | 1 month of prednisone, then 6 had LAM and 1 had IFN-α (n = 7) | ACEI, fish oil, or neither (n = 10) | NA | 5–120 mo | 3 | ① |
| Sun et al[21] | 2012 | Seoul, Korea | CCT | 37 | M:9 F:1 | Antiviral drugs (n = 6) | ACEI or ARB (n = 4) | NA | mean 87 mo (8–187) | 1 | ① |
| Lai et al[22] | 1991 | Hong Kong, China | CCT | 30 | M:14 F:2 | IFN-α2b (3 million units, 3 times/week, subcutaneous injection)(n = 5) | diuretic agents/ dipyridamole /no treatment (n = 11) | 12 weeks | 55.1 mo | 0 | ①② |
| Lin et al[23] | 1995 | Taiwan, China | RCT | 6.5±3.3 | M:29 F:11 | IFN-α2b (5μ if body weight<20 kg, 8μ if body weight>20 kg, 3 times/week subcutaneous injection) | control of edema and hypertension | 12 mo | 24 mo | 0 | |

Abbreviations: RCT, randomized controlled trial; CCT, cohort clinical trial; M, male; F, female; LAM, lamivudine; ACEI, angiotensin converting enzyme inhibitors; ARB, angiotensin receptor blocker; NA, not available; mo, months; ①Renal remission; ②eGFR (estimated glomerular filtration rate); ③HBeAg clearance.

included only one patient treated with IFNs, therefore, because groups could not be formed with only one patient, we removed this data when the statistical analyses were performed. Proteinuria was significantly decreased in the NA group (OR = 6.67, 95% CI: 2.50–17.80) and the IFN group (OR = 38.76, 95% CI: 7.03–213.71, Fig 3). Heterogeneity, calculated using the $I^2$ statistic with a fixed-effect model, was $I^2$ = 0%, $P$ = 0.58 in the IFN group, and $I^2$ = 1%, $P$ = 0.39 in the NA group.

**3.3 Subgroup analysis of the association between efficacy and age.** Six studies included adult patients, and the other two studies included pediatric patients. The adult group had 103 patients, composing 56.6% of the total 182 patients, and the pediatric group had 79 patients, accounting for 43.4%. A fixed-effect model was used. The proteinuria evaluation in both the pediatric patients (OR = 57.71, 95% CI: 7.21–461.82) and the adult patients (OR = 6.38, 95% CI: 2.51–16.24) emphasized the good effect of antiviral therapy on adult patients and pediatric patients (Fig 4). There was no heterogeneity in the CR rate in trials with pediatric patients ($I^2$ =

**Table 2. Newcastle–Ottawa scale for the observational studies.**

| Study | Year | Selection (up to 4) | Comparability (up to 2) | Outcome (up to 3) |
|---|---|---|---|---|
| Sun et al.[17] | 2017 | 3 | 1 | 2 |
| Bhimma et al.[18] | 2002 | 3 | 1 | 2 |
| Tang et al.[19] | 2005 | 3 | 1 | 3 |
| Panomsak et al.[20] | 2006 | 3 | 0 | 2 |
| Sun et al.[21] | 2012 | 3 | 1 | 3 |
| Lai et al.[22] | 1991 | 2 | 1 | 3 |

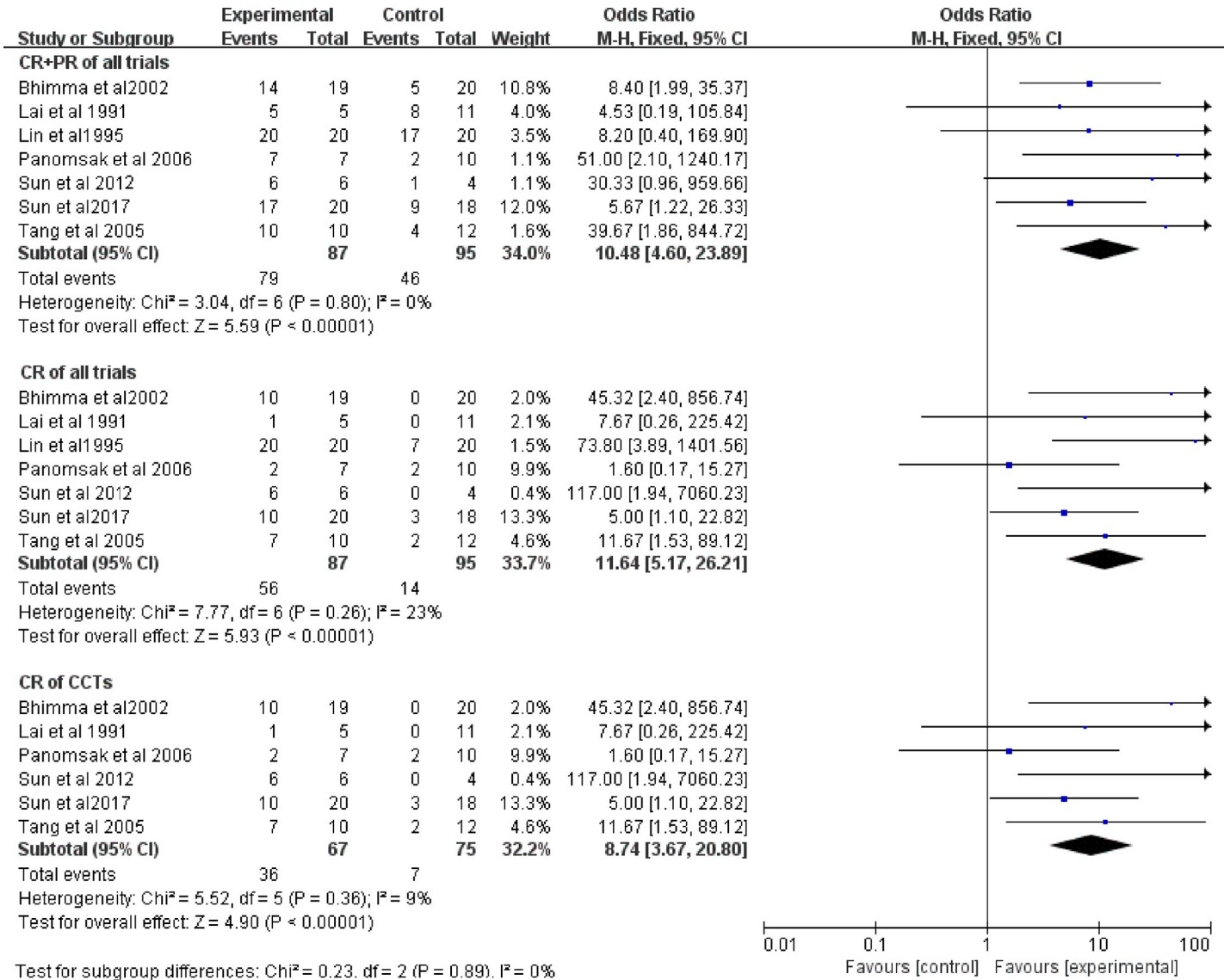

**Fig 2. CR and CR+PR with antiviral therapy in all trials and CCTs.** OR: odds ratio; CR: complete remission; PR: partial remission.

0%, $P = 0.82$) or adult patients ($I^2 = 0\%$, $P = 0.43$), which shows the relationship between age and heterogeneity.

**3.4 Subgroup analysis of groups at the 12-month follow-up.** Three trials (n = 100) mentioned the proteinuria remission rate at the 12-month follow-up, and the results showed that the CR (OR = 12.89, 95% CI: 1.56–106.41) of proteinuria was obviously higher in the antiviral treatment group than in the control group. Heterogeneity using the $I^2$ statistic with a random effect model was $I^2 = 69\%$, $P = 0.04$, and the test for subgroup difference was $P = 0.86$ (Fig 5).

## 4. Effects on the eGFR

The renal function of patients was observed in four of the seven trials during the follow-up. Anti-viral therapy did not affect the eGFR (MD = 5.74, 95% CI: -4.24–15.73), and the heterogeneity was $I^2 = 44\%$ with a fixed-effect model (Fig 6). The sensitivity analysis revealed that heterogeneity was mainly impacted by the studies by Sun's study[17] (S4 Fig).

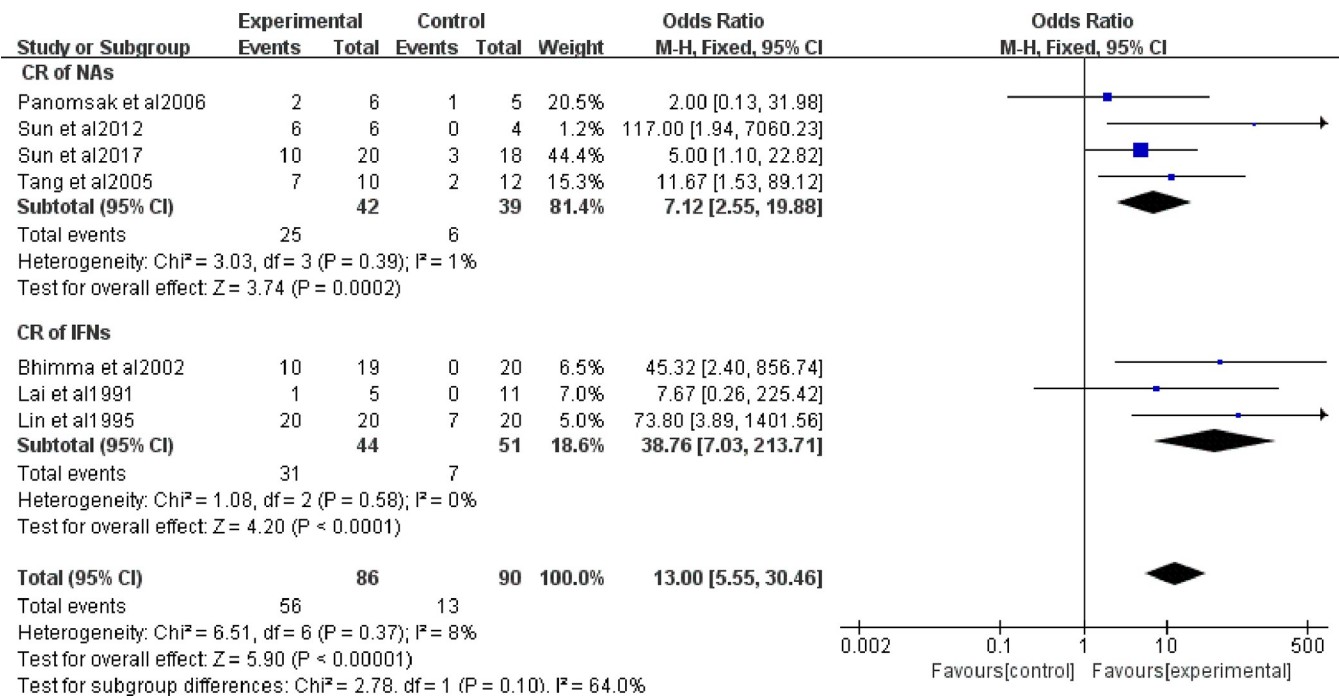

**Fig 3. CR and CR+PR with IFNs and NAs in HBV-GN patients.** OR: odds ratio; CR: complete remission.

**Fig 4. CR with antiviral therapy in adult patients and pediatric patients.** OR: odds ratio; CR: complete remission.

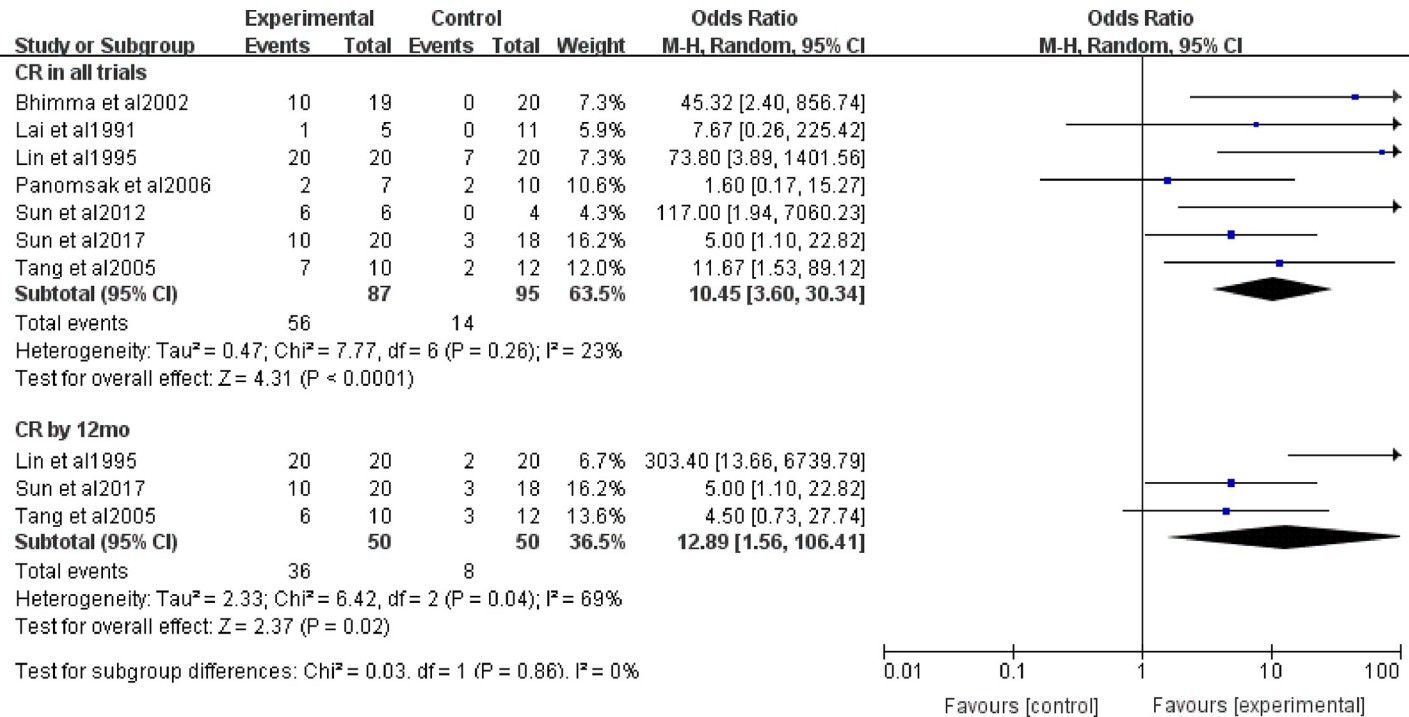

**Fig 5. CR and CR+PR with at the 12-month follow-up.** OR: odds ratio; CR: complete remission; PR: partial remission.

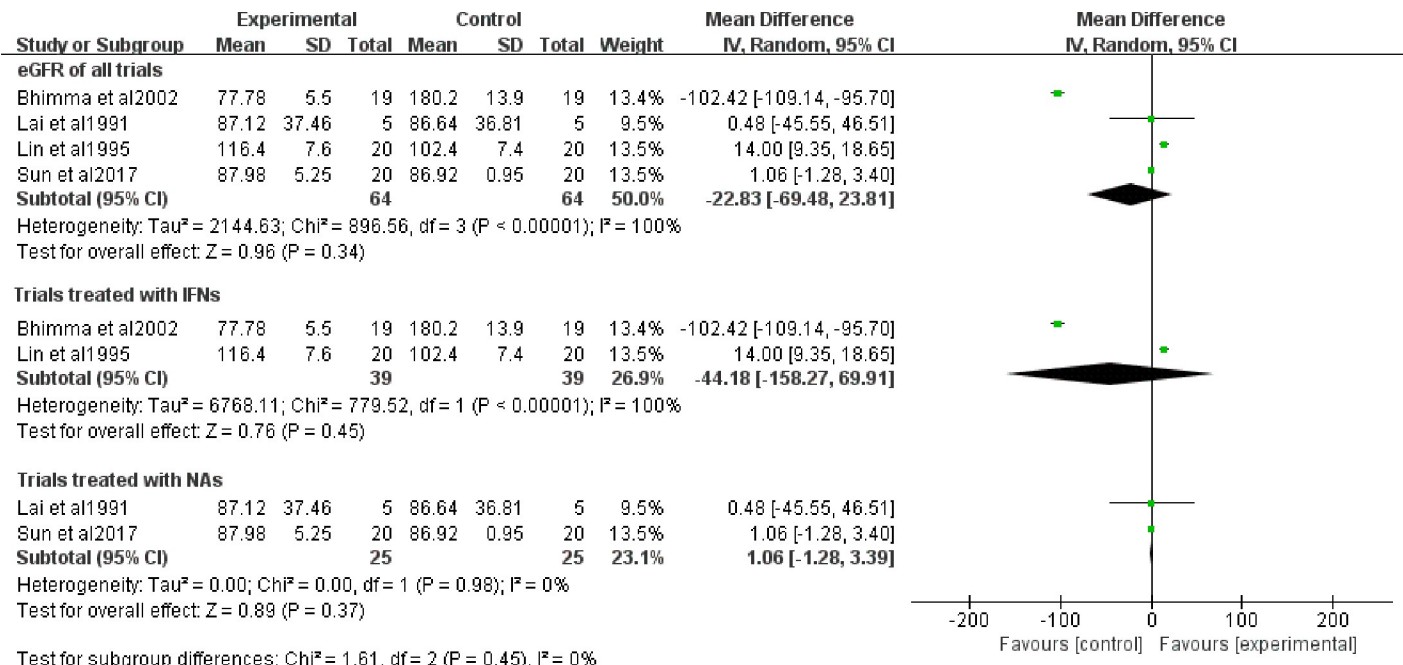

**Fig 6. eGFR in antiviral therapy.** OR: odds ratio.

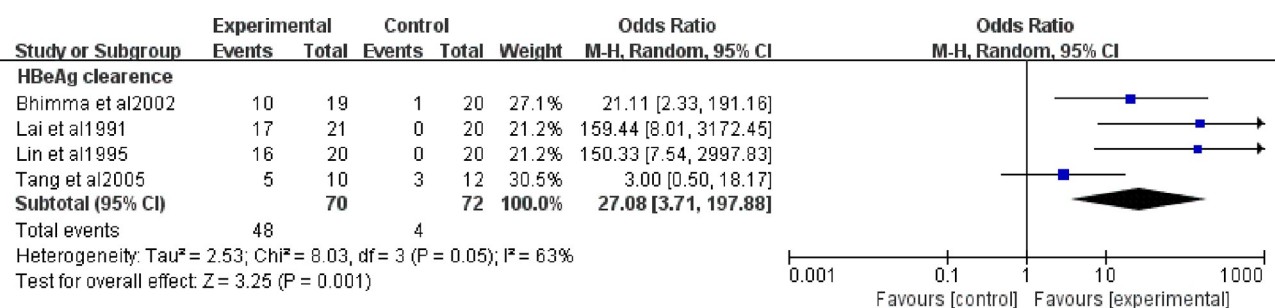

**Fig 7. HBeAg clearence in antiviral therapy.** OR: odds ratio.

## 5. Clearance of HBeAg in antiviral therapy

Four trials, including 142 cases, investigated the impact of antiviral therapy on HBeAg clearance in HBV-GN patients. We evaluated HBeAg clearance after antiviral therapy, and the heterogeneity using the $I^2$ statistic was 63%, $P = 0.05$. The test for the overall effect was $P<0.00001$ with a random effect model (Fig 7). HBeAg significantly decreased after antiviral therapy (OR = 27.08, 95% CI: 3.71–197.88). A sensitivity analysis was performed and showed that the study by Tang's study[19] was the main factor impacting the results. (S5 Fig)

## 6. Publication bias

Publication bias may exist when no significant findings remain unpublished thus artificially inflating the apparent magnitude of an effect. Begg's test and Egger's test were performed to assess the publication bias for total remission of proteinuria and suggested the absence of publication bias (Begg's test: $P = 0.230$; Egger's test: $P = 0.191$).

## 7. Adverse events

Six trials mentioned side effects during treatment, including 201 patients. Three studies were treated with IFNs (n = 95), and adverse events were encountered after approximately 3 months of therapy and were mainly flu-like illnesses and pains. Later, side effects appeared after 6 months of therapy were mainly psychiatric problems. The most common events were flu-like syndrome (50/95, 52.6%), fever (26/95, 27.3%), fatigue (27/95, 28.4%) and various kinds of pains (46/95, 48.2%), including myalgia (n = 30), headaches (n = 28), abdominal pain (n = 2) and arthralgia (n = 8). The above symptoms were not serious and subsided spontaneously or were relieved by analgesics. The psychiatric symptoms (21/95, 22.1%), such as anxiety and loss of interest (n = 6), insomnia (n = 8), depression (n = 4) and suicidal ideation (n = 3). These side effects disappeared quickly after reducing the dose of IFNs. Other symptoms were rare including anorexia, nausea, chills, neutropenia and thrombocytopenia. Compared with IFNs, the side effects in patients treated with NAs (n = 70) were fewer, and the adverse events observed in the remaining three trials seemed to be random with no specific symptoms mentioned.

## Discussion

Our meta-analysis showed that most patients with HBV-GN were successfully treated with antiviral therapy, and the summary estimate for the OR of proteinuria decrease at the end of therapy was 11.64 (95% CI: 5.17–26.21). Similar results were reported in previous meta analyses[4,6,7,9,10]. Compared with CR, the quantity of PR could not reflect the treatment effects,

so our study focused on total remission (CR+PR) and CR. From the numerical results, we found that the included patients all experienced benefits in total remission and CR. However, the CR statistics showed larger heterogeneity than did total remission. This finding indicated that the clinical efficacy of these trials differed. To make accurate decisions regarding treatment for different groups of patients and explore the sources of heterogeneity, we carried out subgroup and sensitivity analyses to identify the influencing factors. In the prespecified subgroup analysis, the test for differences among age subgroups showed a moderate difference with $P$ = 0.06, and the OR for pediatric patients was 57.71, which was much larger than that of adult patients (OR = 6.38). Heterogeneity decreased to $I^2$ = 0% in both pediatric and adult patients. The results showed that pediatric patients were more sensitive to antiviral therapy than adults, which illustrated that children recovered more easily. Subgroup differences in drug type ($P$ = 0.10) were not very significant. IFNs were more effective than NAs. This result was not inconsistent with Yang's study[10], which proved IFNs and NAs were equally effective at causing proteinuria remission. However, Yang's study[10] only focused on MN (membranous nephropathy) patients, we considered different research subjects might lead to conflicting conclusions. Furthermore, the study type and follow-up duration subgroups did not differ in terms of results. The sensitivity analysis showed that Sun's study[17] and Panomsak's study [20] were the main source of influence on CR. Renal pathological examination in these studies revealed that the pathologic types were various, IgA nephropathy had a lager proportion. While, almost all of the HBV-GN patients presented MN in other studies, except 2 patients in Bhimma's study[18] were MPGN (Mesangial Proliferative Glomerular Nephritis), which may be the reason for influencing heterogeneity.

The effect of antiviral therapy on eGFR was not obvious. The first reason for poor efficacy was age because the standards for eGFR for adult patients and pediatric patients are not the same, so they are not suitable for combining together for analysis. Second, body weight and sex could affect eGFR, and the patients that were included may have been heterogeneous with respect to these indicators. In addition, antiviral drugs associated with nephrotoxicity, such as adefovir dipivoxil, tenofovir and IFNs, should also be taken into consideration. These are excreted through the kidney, and high concentrations of both the drugs and their metabolites can cause renal injury. Izzedine's study[24] demonstrated that adefovir dipivoxil (120 mg/d) could increase the serum creatinine level by 44 μmol/L in HIV-infected patients. Kara AV's study[25] found that tenofovir could decrease the eGFR and that the change in eGFR was not statistically significant. The after sensitivity analysis also confirmed that Sun's study[17], which chose lamivudine as an intervention, has a great influence on the heterogeneity. In addition, inconsistency of renal pathologic types may also be a factor. Wang et al[9] performed a meta-analysis and decleared that NA monotherapy could protect renal function and preventing Scr (Serum creatinine) elevation. However, our results didn't show significant improvement in renal function, especially under treatment of NAs.

Our results showed that antiviral therapy could increase HBeAg seroconversion. Other meta analyses also showed the same results[6,7,9,10]. HBeAg seroconversion marks the immune-active phase of disease transition to the inactive carrier state[26]. Among the included trials, Tang's study[19] made the results heterogeneous. The sample size of their trial was smaller, and we considered that the results were more likely to be biased. Serum HBV DNA is a sensitive quantitative index that could facilitate the monitoring of response to treatment. The National Institutes of Health Workshop proposed an arbitrary serum HBV DNA level of $10^5$ copies/mL to differentiate chronic hepatitis B from an inactive carrier state. Another study claimed that 33% of patients with HBeAg-negative chronic hepatitis had HBV DNA levels that were persistently above $10^5$ copies/mL[27,28]. According to these data, we considered that in terms of reflecting the disease condition, HBeAg clearance could replace

serum HBV DNA to some extent. Two trials[18,19] included in our study mentioned serum HBV DNA as an outcome indicator, but they did not provide concrete values; they only described the numbers of patients that had a decrease in serum HBV DNA. For the above reasons, we did not evaluate the effect of drugs based on this indicator.

The incidence of adverse reactions for IFNs was greater than for NAs. The side effects of the groups were different from each other, but most of them were within the tolerable range. Reducing the dose could result in the relief of symptoms.

Antiviral drugs were initially intended to treat HBV infection, but the present findings proved these have a curative four major mechanisms: (1) HB immune complexes of macromolecules were deposited in the mesangial area and sub-endodermis, leading to different pathological types. One is circulating immune complexes of viral antigen and host antibody, and the other is in situ immune complexes involving viral antigens bound to glomerular structures [29]. Previous studies have reported that HBeAg is the primary antigen related to subepithelial deposits in patients with HBV-MN[30,31]. Takekoshi et al[32] supported the view that immune complexes, especially those containing HBeAg, were essential in the pathogenesis of HBV-GN.

(2) HBV infection of nephrocytes induces a cytopathic effect[33]. Thus, the hypothesis does not have a definite answer and needs further evidence.

(3) Virus-induced specific immunological effector mechanisms (specific T lymphocyte or antibody) that damage the kidney, which induces pathologic damage (such as cell apoptosis and proliferation) and HBV replication in renal tissue. Host immunodeficiency and virus variation are thought to be the most subtle contributions to persistent HBV infection.

(4) HBV in tissue could express viral proteins or pro-inflammatory cytokines, resulting in renal injury[34].

Human HBV is a small, enveloped, primarily hepatotropic DNA virus[35]. NAs can suppress viral replication by inhibiting the activity of HBV polymerase in the infected liver, but they cannot eliminate existing viruses or eradicate HBV cccDNA[36]. IFNs bind to cellular receptors and induce large numbers of genes termed IFN-stimulated genes (ISGs). ISGs are multifunctional and can inhibit HBV replication at different stages: they can suppress cccDNA transcription and accelerate cccDNA decay, inhibit the activity of HBV enhancers, influence posttranscriptional control of HBV replication, and inhibit HBV nucleocapsid formation. Furthermore, IFNs can also activate antiviral enzymes and stimulate an immunomodulatory effect[37].

In summary, pediatric patients were sensitive to antiviral drugs, which could be used fairly easily to mitigate proteinuria, and they required a shorter duration of treatment than did adults. Therefore, we think that further implementation of randomized controlled trials on pediatric patients would be straightforward. A comparison between IFNs and NAs demonstrated that IFNs were more effective, but the incidence of side effects was higher for IFNs than for NAs. We conclude that an individualized therapy program considering treatment efficacy and side effects is the best option for patients.

Several limitations in our meta-analysis should be considered. First, the quality of our analysis could be improved by a larger sample size and longer follow-up. Observational studies are not the best study design to answer intervention questions because of bias and confounding, the effects of which are unpredictable and may result in an overestimate, an underestimate, or true estimate of the effect. The limited number of studies available and patients enrolled in each study reduced the strength of our final conclusions. Second, dosage and treatment duration for drug administration and the follow-up duration might have influenced the clinical trial data. Third, patients enrolled in our study had a variety of renal lesion types. Because the clinical results of patients with renal puncture pathology are unavailable, no further subgroup

analysis could be performed. Finally, trials in our meta-analysis were all in English and conducted in Asian countries and mostly limited to China. This geographic concentration is probably the reason for the high prevalence of HBV infection in these countries. As a result, our conclusions concerning the efficacy and safety of antiviral treatments may not apply to European or American patients.

## Conclusion

In summary, some broad conclusions can be drawn. Our results strengthen the evidence that antiviral therapy is effective in antiviral treatment groups compared to control groups. It was particularly effective in attaining remission of proteinuria and improving HBeAg clearance but failed to improve the eGFR. Antiviral treatment has a definite therapeutic effect on children, and IFNs seem more effective than NAs. However, NAs showed fewer adverse reactions than IFNs, so we tend to choose drugs based on a patient's specific condition. Follow-up duration and study type did not have major impacts on the results. Our study highlights the need for randomized controlled trials in the area of HBV-GN, so we can evaluate the best antiviral approach.

## Supporting information

**S1 Fig. Search strategy for the PubMed database.**
(TIF)

**S2 Fig. The Cochrane Collaboration's tool for randomized controlled trials.**
(TIF)

**S3 Fig. Sensitivity analysis of CR in all trials.**
(TIF)

**S4 Fig. Sensitivity analysis of the eGFR in all trials.**
(TIF)

**S5 Fig. Sensitivity analysis of HBeAg clearance in all trials.**
(TIF)

**S1 Table. PRISMA 2009 checklist.**
(DOC)

## Author Contributions

**Data curation:** Ge Li.

**Formal analysis:** Ge Li.

**Project administration:** Hongtao Yang.

**Resources:** Hongtao Yang.

**Supervision:** Tong Ren, Hongtao Yang.

**Validation:** Shouci Hu.

**Visualization:** Baohui Fu, Yue Ji.

**Writing – original draft:** Baohui Fu, Yue Ji.

**Writing – review & editing:** Baohui Fu, Yue Ji, Shouci Hu, Maheshkumar Satishkumar Bhuva.

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
