## [Decision Letter · Decision Letter 0]

27 Aug 2019

PONE-D-19-20164

Efficacy and safety of anti-viral therapy for Hepatitis B virus-associated glomerulonephritis: a meta-analysis

PLOS ONE

Dear Dr Yang,

Thank you for submitting your manuscript to PLOS ONE. After careful consideration, we feel that it has merit but does not fully meet PLOS ONE’s publication criteria as it currently stands. Therefore, we invite you to submit a revised version of the manuscript that addresses the points raised during the review process.

This interesting meta-analysis was evaluated by the reviewers and the feedback was positive regarding the acceptance of their manuscript in PLOS ONE. Reviewers asked for minor comments which should be considered by the authors in the revised manuscript. One of the reviewers raised a major concern regarding the "additional outcome measure of suppression or clearance of hepatitis B DNA" which should be carefully take into consideration by the authors. 

We would appreciate receiving your revised manuscript by Oct 11 2019 11:59PM. To enhance the reproducibility of your results, we recommend that if applicable you deposit your laboratory protocols in protocols.io, where a protocol can be assigned its own identifier (DOI) such that it can be cited independently in the future. For instructions see: http://journals.plos.org/plosone/s/submission-guidelines#loc-laboratory-protocols

We look forward to receiving your revised manuscript.

Kind regards,

Heidar Sharafi

Academic Editor

PLOS ONE

Journal Requirements:

Additional Editor Comments:

In my evaluation, I found the study well-designed and -presented and can have great contribution in literature and clinical management of patients however, I found some additional points to be corrected for improvement of manuscript presentation. Please find them:

1. Authors used # to show the authors with equal contribution however, only one author's name tagged with #.

2. In the first line of abstract it has been pointed to "corticosteroid" while there is nothing on corticosteroids in the study design and results.

3. In abstract, "proteinuria 18 (OR=10.48, 95% CI: 4.60�23.89, I2=0%, P=0.80)". Is the P for heterogeneity or overall effect? It should be clarified.

4. I guess the results for eGFR should be the mean difference of eGFR instead of OR.

5. In introduction, "where four studies[4,6,7]" while 3 study have been cited.

6.  In Methods, "In the absence of statistically significant heterogeneity I2>50%" I2 >50% means significant heterogeneity.

7. Page 11, Line 14, "Eight studies were conducted evaluating" 7 or 8?

8. Page 12, Line 1, "fixed model" or fixed-effect model?

9. Page 12, Line 4, heterogeneity was highly significant (I2=0%, P=0.80). I2=0% and significant heterogeneity?

10. Page 12, Line 10, "We conducted a sensitivity analysis of all studies (Fig 5)." but 6 studies entered in sensitivity analysis.

11. Page 12, Lines 12-15 and in Figure 4, these overall effects caused by repeated inclusion of studies in the meta-analysis. I recommend to exclude it from manuscript and from the Total OR (95% CI) in the bottom part of forest plot in Figure 4. The same can be true for Figure 8 and Figure 9 with inclusion of studies multiple times make a false total overall effect.

12. Page 14, "Meanwhile, in trials with IFNs, the treatment group was not obviously affected (P<0.00001, I2=100%). Are they talking about heterogeneity? The heterogeneity is high with I2=100.

13. For pooled effect of clearance of HBeAg in antiviral therapy, the I2=63% and yet using fixed effect model.

14. My main concern is the discussion. It is currently concentrated on the results and less discuss the findings. There are much talked about heterogeneity while it is a technical issue while they should concentrate on overall effect of the pooled analysis. They added some information on the mechanism of HBV-GN, I am not against presentation of such information however much is not needed and there is lack of information on how significant is the treatment of HBV-GN and what is the recommendation of authors for treatment of HBV-GN based on the results of meta-analysis. Moreover, in a paragraph, the limitations have been included however, I would like to ask authors to highlight what are the strengths of their study and how it is changing the current knowledge while few meta-analysis have been published even recently (2016).

15. In PRISMA flow diagram, "115 records after duplicates removed". Are these 115 were duplicates excluded? It is really confusing in the current drawing.

16. There is an additional unnecessary Total effect in Figure 11 can be excluded.

Reviewers' comments:

Reviewer's Responses to Questions

**Comments to the Author**

1. Is the manuscript technically sound, and do the data support the conclusions?

Reviewer #1: Yes

Reviewer #2: Yes

Reviewer #3: Yes

2. Has the statistical analysis been performed appropriately and rigorously? 

Reviewer #1: Yes

Reviewer #2: Yes

Reviewer #3: Yes

3. Have the authors made all data underlying the findings in their manuscript fully available?

Reviewer #1: Yes

Reviewer #2: Yes

Reviewer #3: Yes

4. Is the manuscript presented in an intelligible fashion and written in standard English?

Reviewer #1: Yes

Reviewer #2: Yes

Reviewer #3: Yes

5. Review Comments to the Author

Reviewer #1: A number of sections in this document need to be reviewed and improved English syntax and lexicon is advised

Wording: Page 8 line 11 should be changed from "primary aftermath measures" to "primary outcome measures"

There should be an additional outcome measure of suppression or clearance of hepatitis B DNA\\

The current population estimate on page 9 line 9 is there are 292 million people with Hepatitis B today world health organization and Polaris

The authors need to highlight that there are two types of tenofovir: TDF and TAF

tenofovir is no capitalized

The letter B should be capitalized throughout the manuscript when discussing the HBV or hepatitis B

glomerulonephritis does not need to be capitalized

adefovir and tenofovir analogues are known nephrotoxins

Did the authors look at nephrotoxicity risk in these manuscripts? And effect on GFR ? this may partially explain the lack of change in GFR?

Reviewer #2: Dear Editor

Thanks for inviting me to review the manuscript PONE-D-19-20164 entitled "Efficacy and safety of anti-viral therapy for Hepatitis B virus-associated glomerulonephritis: a meta-analysis". The subject is novel with a correct methodology, and its manuscript is well-written. My minor comments to the authors are as follow:

- There are some minor writing mistakes in the manuscript. (HVB-GN >>> HBV-GN, Mesh >>> MeSH, and...) Please read the manuscript carefully to find and correct them.

- There is no need to use thirteen figures to show the results. You can remove some unnecessary figures or move them into supplements section. Figure one in unnecessary you can move it to the supplements. Showing the publication bias in a figure is not necessary.

- There is no need to acknowledge the language editor institute.

Regards

Reviewer

Reviewer #3: This is a well-written paper out of good work. Although the size of the data is not large, it reaffirms previous information on the efficacy of anti-HBV treatments on HBV-GN.

6. PLOS authors have the option to publish the peer review history of their article (what does this mean?). If published, this will include your full peer review and any attached files.

Reviewer #1: Yes: Robert G Gish

Reviewer #2: Yes: Hamidreza Karimi-Sari

Reviewer #3: Yes: Amir Ali Sohrabpour

---

## [Author Response · Author response to Decision Letter 0]

9 Oct 2019

Dear Editors and reviewers：

Thank you for your letter of comments and for the referee’s comments concerning our manuscript entitled “Efficacy and safety of anti-viral therapy for Hepatitis B virus-associated glomerulonephritis: a meta-analysis”. We have studied all comments carefully and have made corrections. We have send the revised manuscript and revised portion were marked in red. We really hope to meet with your approval.

---

## [Decision Letter · Decision Letter 1]

7 Nov 2019

PONE-D-19-20164R1

Efficacy and safety of anti-viral therapy for Hepatitis B virus-associated glomerulonephritis: a meta-analysis

PLOS ONE

Dear Dr Yang,

Thank you for submitting your manuscript to PLOS ONE. After careful consideration, we feel that it has merit but does not fully meet PLOS ONE’s publication criteria as it currently stands. Therefore, we invite you to submit a revised version of the manuscript that addresses the points raised during the review process.

After initial evaluation of revised manuscript, I decided to send the manuscript for peer-review again to improve its scientific presentation to reach a level aligned for publication in PLOS ONE. One of the main points is proofing the English expression of manuscript before acceptance. Authors can ask a colleague expert in English writing or a professional English editing service to improve their manuscript English expression.

We would appreciate receiving your revised manuscript by Dec 22 2019 11:59PM. To enhance the reproducibility of your results, we recommend that if applicable you deposit your laboratory protocols in protocols.io, where a protocol can be assigned its own identifier (DOI) such that it can be cited independently in the future. For instructions see: http://journals.plos.org/plosone/s/submission-guidelines#loc-laboratory-protocols

We look forward to receiving your revised manuscript.

Kind regards,

Heidar Sharafi

Academic Editor

PLOS ONE

Additional Editor Comments (if provided):

I have few minor comments on the revised manuscript:

1. Conclusion of abstract should be concise.

2. Page 3, Line 18, use "nucleos(t)ide analogs (NAs)" instead of "nucleoside analogs (NAs)".

3. I didn't find S1 Table.

4. It was stated "Heterogeneity between data sources was noted in the CR rate of all trials (I2=23%, P=0.26) and CCTs (I2=9%, P=0.36)." while based on I2 there is no significant heterogeneity between studies. Page 12, Lines 9-11

5. What is "(I2=0%<50%, P=0.77)"? Page 12, Line 14

6. Page 12, Lines 11-12, "Our analysis determined that the study by Lin et al[17] was the source of heterogeneity.", based on Figure 4, I expect the study by Fang et al to cause heterogeneity.

7. Figure 3, the scale of first plot and the two next plots should be the same. all 0.001-1000 or 0.01-100.

8. Regarding the sensitivity analysis plots, why the scales are not aligned with the results of meta-analysis? For example, in figure 4, the OR is 2.58 while I expect an OR 11.64 based on the meta-analysis. This is true for other sensitivity analysis plots as well.

9. Figure 7, in caption of plot, the OR>1 should favours experimental and OR<1 should favours control.

Reviewers' comments:

Reviewer's Responses to Questions

**Comments to the Author**

1. If the authors have adequately addressed your comments raised in a previous round of review and you feel that this manuscript is now acceptable for publication, you may indicate that here to bypass the “Comments to the Author” section, enter your conflict of interest statement in the “Confidential to Editor” section, and submit your "Accept" recommendation.

Reviewer #2: All comments have been addressed

Reviewer #4: (No Response)

2. Is the manuscript technically sound, and do the data support the conclusions?

Reviewer #2: Yes

Reviewer #4: Yes

3. Has the statistical analysis been performed appropriately and rigorously? 

Reviewer #2: Yes

Reviewer #4: Yes

4. Have the authors made all data underlying the findings in their manuscript fully available?

Reviewer #2: Yes

Reviewer #4: No

5. Is the manuscript presented in an intelligible fashion and written in standard English?

Reviewer #2: No

Reviewer #4: No

6. Review Comments to the Author

Reviewer #2: Thanks for revising your work. Almost all of my comments are addresses, but a minor issue is still remained in the manuscript. Eleven figures are not recommended in a paper. You can move 4 other figures (2, 4, 9, 11) to the supplementary materials (supporting information).

Regards

Reviewer

Reviewer #4: Dear Associated Editor

I am very thankful for giving me the opportunity to review the manuscript entitled “Efficacy and safety of anti-viral therapy for Hepatitis B virus-associated glomerulonephritis: a meta-analysis” and evaluate it for publishing in PLOS ONE. This manuscript has been revised once. I appreciate the efforts authors put into this work. However, I would like to mention some major point for authors:

1. The manuscript needs writing edit for English language:

a) Improve overall readability

b) Use shorter sentences

c) Pay attention to punctuation (for instance in the third line of introduction dot and comma used together.)

d) There are several grammatical errors in the manuscript (As an example the first sentence of the second paragraph of introduction has totally wrong structure)

2. Authors declared that there were several meta-analysis in this field and there were published before 2011 and that is the reason they performed this study. Only two included studies published after 2011, do they have great impact on the results and evidences?! What is the benefit of your study?

3. Mention the exact PICOS of the study at the end of the introduction section.

4. Mention the exact time you performed your search, in the methods section.

5. Provide search strategy for at least one database.

6. Why did you searched PUBMED and MEDLINE?? When you are searching PUBMED you are searching MEDLINE, too.

7. Provide more information about covering grey literature. How did you cover this and which conferences did you check.

8. Did you consider any specific language in the search? Four of seven included studies are from China.

9. Write “Eligibility criteria” section in an organized way. It is really confusing.

10. Mention the outcome you expected to find in the included articles, in the “Eligibility criteria” section.

11. In “Data extraction” section, mention the exact variables you extracted from the studies. Do not use “So on”.

12. Did you pay attention to funding source of the studies? You did not mention this.

13. Third line of “Statistical analysis” section, the sentence “In the absence of...” can be deleted. You have discussed this further.

14. You included seven studies. Four from China, two from other parts of Asia and one from Africa. How do you discuss this distribution?

15. “Subgroup analysis of association between efficacy and study types” part should be totally revised. First, discuss the heterogeneity results and then report the analysis results.

16. In “Subgroup analysis of association between efficacy and study types” section, heterogeneity of CCTs was reported twice in the 8th and 11th line with different statistics. Please rewrite this part and make it clear if they are different variables.

17. In the second line of “Effects on eGFR” section, replace “anti-virus therapy” with “anti-viral therapy”.

18. In the fifth line of “Effects on eGFR” section, authors declared “it seems that NAs has therapeutic effect on eGFR”. However, the reported confidence interval includes zero. Please clarify this part.

19. The discussion part is written like introduction section. Study results should be discussed and compared with other studies. Discussion section needs an overall revision.

7. PLOS authors have the option to publish the peer review history of their article (what does this mean?). If published, this will include your full peer review and any attached files.

Reviewer #2: No

Reviewer #4: Yes: Maryam Tajik

---

## [Author Response · Author response to Decision Letter 1]

20 Dec 2019

Thank you for your kind comments on our manuscript. We have carefully revised the manuscript according to the reviewer’s comments. Based on the suggestions, we have made an extensive modification on the revised manuscript.. The changes to our manuscript within the document were aslo highlighted by using red colored text.

---

## [Editor Report · Decision Letter 2]

23 Dec 2019

Efficacy and safety of anti-viral therapy for Hepatitis B virus-associated glomerulonephritis: a meta-analysis

PONE-D-19-20164R2

Dear Dr. Yang,

We are pleased to inform you that your manuscript has been judged scientifically suitable for publication and will be formally accepted for publication once it complies with all outstanding technical requirements.

With kind regards,

Heidar Sharafi

Academic Editor

PLOS ONE
---

## [Editor Report · Acceptance letter]

2 Jan 2020

PONE-D-19-20164R2 

Efficacy and safety of anti-viral therapy for Hepatitis B virus-associated glomerulonephritis: a meta-analysis 

Dear Dr. Yang:

I am pleased to inform you that your manuscript has been deemed suitable for publication in PLOS ONE. Congratulations! Your manuscript is now with our production department. 

With kind regards,

on behalf of

Dr. Heidar Sharafi 

Academic Editor

PLOS ONE